# The Use of Natural Language Processing for Computer-Aided Diagnostics and Monitoring of Body Image Perception in Patients with Cancers

**DOI:** 10.3390/cancers15225437

**Published:** 2023-11-16

**Authors:** Elwira Gliwska, Klaudia Barańska, Stella Maćkowska, Agnieszka Różańska, Adrianna Sobol, Dominik Spinczyk

**Affiliations:** 1Department of Food Market and Consumer Research, Institute of Human Nutrition Sciences, Warsaw University of Life Sciences (WULS-SGGW), 159C Nowoursynowska Street, 02-776 Warsaw, Poland; elwira.gliwska@pib-nio.pl; 2Cancer Epidemiology and Primary Prevention Department, The Maria Sklodowska-Curie National Research Institute of Oncology, 15B Wawelska Street, 02-034 Warsaw, Poland; 3Department of Medical Informatics and Artificial Intelligence, Faculty of Biomedical Engineering, Silesian University of Technology, Roosevelta 40, 41-800 Zabrze, Poland; klaudia.baranska@polsl.pl (K.B.); stella.mackowska@polsl.pl (S.M.);; 4Polish National Cancer Registry, Maria Sklodowska-Curie National Research Institute of Oncology, 02-781 Warsaw, Poland; 5Department of Oncological Propaedeutics, Medical University of Warsaw, 00-518 Warsaw, Poland

**Keywords:** body image perception, computer-aided diagnostics, computer-aided diagnostics and monitoring, natural language processing, head and neck cancer, upper gastrointestinal tract cancer

## Abstract

**Simple Summary:**

Psychological assessment of a cancer patient is a challenge due to the difficulty of the issues raised and, additionally, the lack of a sufficient number of psycho-oncologists. The article proposes a minimally invasive automatic method of this assessment using a note prepared by the patient about his body image. The proposed method allows you to determine the general attitude, the intensity of the five basic emotions (happiness, fear, sadness, anger, disgust), and potential areas of difficulty in terms of: body image, self-esteem, and acceptance of the environment. The combination of the attributes of patient’s condition determined by the method can indicate the direction of support for the patient, relatives, direct medical personnel, and psycho-oncologists.

**Abstract:**

Background: Head and neck cancers (H&NCs) constitute a significant part of all cancer cases. H&NC patients experience unintentional weight loss, poor nutritional status, or speech disorders. Medical interventions affect appearance and interfere with patients’ self-perception of their bodies. Psychological consultations are not affordable due to limited time. Methods: We used NLP to analyze the basic emotion intensity, sentiment about one’s body, characteristic vocabulary, and potential areas of difficulty in free notes. The emotion intensity research uses the extended NAWL dictionary developed using word embedding. The sentiment analysis used a hybrid approach: a sentiment dictionary and a deep recursive network. The part-of-speech tagging and domain rules defined by a psycho-oncologist determine the distinct language traits. Potential areas of difficulty were analyzed using the dictionaries method with word polarity to define a given area and the presentation of a note using bag-of-words. Here, we applied the LSA method using SVD to reduce dimensionality. A total of 50 cancer patients requiring enteral nutrition participated in the study. Results: The results confirmed the complexity of emotions in patients with H&NC in relation to their body image. A negative attitude towards body image was detected in most of the patients. The method presented in the study appeared to be effective in assessing body image perception disturbances, but it cannot be used as the sole indicator of body image perception issues. Limitations: The main problem in the research was the fairly wide age range of participants, which explains the potential diversity of vocabulary. Conclusions: The combination of the attributes of a patient’s condition, possible to determine using the method for a specific patient, can indicate the direction of support for the patient, relatives, direct medical personnel, and psycho-oncologists.

## 1. Introduction

Cancer is one of the leading causes of death worldwide, responsible for nearly 10 million deaths in 2020. Head and neck cancers (H&NCs) constitute a significant part of all cancer cases (“Worldwide cancer data|World Cancer Research Fund International”, n.d.) [1]. Currently, we expect an increase in the number of H&NCs, and thus, a greater number of long-term complications will be observed in patients as a result of cancer treatment. Head and neck cancers are often treated with medical interventions that affect appearance and may interfere with patients’ self-perception of their own bodies (Clarke et al., 2014 [2]). The disease process strongly affects the efficiency, mobility, and quality of functioning in everyday life, but long-term complications of treatment are no less important. The problems often affecting H&NC patients are significant weight loss in a short period of time, food intake disorders as a result of anti-cancer therapies, changes in taste and smell, loss of muscle mass and strength, or speech disorders. In addition, cancer treatment procedures in H&NC often interfere with the ability to eat food orally; therefore, patients require feeding directly into the stomach, intestine, or parenterally. An effective and individualized nutritional plan should include not only the appropriate nutritional assessment, but also enteral or parenteral nutrition if needed (Chow et al., 2016 [3]). It has been proven that enteral nutrition positively affects patients’ quality of life (Gliwska et al., 2021 [4]). However, the presence of probes and feeding tubes may be another element affecting the quality of a patient’s life and their body perception and thus their well-being. Due to the significant disorders occurring in the process of oncological treatment in patients with H&NC, it seems important to monitor the way that they perceive themselves and their bodies, which may affect their well-being and motivation.

A comprehensive assessment of the intensity of feeling related to the five main domains of feelings might be a useful part of the psychological diagnostic procedure. Noteworthy, body-perception assessment automation seems to be a promising tool in the work of psychologists and health care specialists by shortening the interview time, improving the efficiency of psychological consultation, and enabling the effective assessment of self-perception of people with speech disorders. Cancer patients are at a higher risk of experiencing mood deterioration and disrupted body image. Therefore, it is essential to assess their body image perception in clinical settings. The objective of this study was to develop an automated tool to evaluate the body image perception of cancer patients. This tool aims to enhance the quality of care in oncological settings by enabling the early detection of body image perception issues and facilitating timely psychological interventions.

### Related Works

Nowadays, the life expectancy of a cancer patient has increased. However, chronic treatment requires long-term therapy and regular check-ups even after recovery, which influences a patient’s life thoroughly (Covrig et al., 2021 [5]). Constant thinking about the disease, frequent medical appointments, patient’s emotional confusion, and external expectations posed by the society lead to psychical discomfort. As The American Cancer Society points out, oncological patients experience lots of traumatic emotional distress from negative cognition and attitudes, which deteriorate various aspects of the patient’s life, such as their work or personal life.

Oncological treatment impacts not only inner organs and physiological functionality but also appearance, which manifests in hair loss, swelling, fatigue, skin deterioration, or loss of body parts (Helms et al., 2008 [6]). As the authors in paper (Rhoten et al., 2013 [7]) indicate, head and neck cancers have a direct and radical influence on patients’ appearance due to the invasive nature of the treatment, which in consequence, disturbs the patient’s body image. As the study (Katz et al., 2000 [8]) claims, body disfigurement, in particular related to the face, may constitute blockades in social communication, achieving success, and in self-identity.

In the literature, we encounter some empirical studies concerning body image among head and neck survivors. The research conducted by Fingeret et al. measured the patient’s pre-treatment body image attitude (Esplen and Fingeret, 2021 [9]). The author used Body Image Scale to assess a patient’s psychological condition in relation to their appearance. The experiment carried out by the authors in the paper (Millsopp et al., 2006 [10]) aimed at identifying the group of patients who mentioned body issues; next, they investigated the clinical factors that might influence the patients’ attitudes, and finally, what treatment the patients were given. The researchers used The University of Washington Quality of Life Scale (UWQOL) in their experiment.

Psychological assessment of a patient’s condition and mood can be supported by tools and techniques of Natural Language Processing. In many cases, such valuation is based on vocabulary analysis of patient’s free notes, which helps diagnose potential areas of psychological distress and shorten the identification process (Bellows et al., 2014 [11]).

Natural Language Processing offers immediate data extraction and analysis of target data. It is used in many fields of medicine, in particular in psychiatry and psychology. NLP methods give promising results in detecting anorexia and support the diagnostic process. They can also be applied in the oncology field. With a rapid increase in the number of people suffering from malnutrition, there is a need for reliable and noninvasive methods that quickly detect mental health problems (López-Úbeda et al., 2019 [12]). Other methods developed by the authors of papers (Rojewska et al., 2022 [13]) intend to create a linguistic profile of a person with anorexia and depict the potential sources of the disease. Other research (Zeng et al., 2021 [14]) focuses on analyzing the clinical structured and unstructured cancer patients records to identify the treatment methods ordinated to the patient and to manage the patient’s medical documentation to tailor the process of further therapy.

The authors propose a method using NLP techniques to determine selected patient attributes. The proposed approach uses dictionary methods, an artificial neural network, and a text topic analysis.

## 2. Materials and Methods

Regarding the need to monitor the patient’s condition mentioned in the introduction and, if possible, a non-invasive method of diagnosis, the authors proposed an automatic method based on Natural Language Processing, including vocabulary analysis, sentiment analysis of one’s body, the ability to determine the intensity of 5 basic feelings, and identify potential areas of difficulty. Figure 1 presents a summary of the elements of the developed method, which are described in more detail below in Section 2.1, Section 2.2, Section 2.3 and Section 2.4.

### 2.1. Text Analysis

This method aims to detect some morphological traits in patients’ notes that may create a linguistic profile of a person with cancer. The analysis includes determining the frequency of using particular parts of speech, such as verbs, verb tense (present or past references), and adjectives (both positive and negative). Moreover, we wanted to detect the intensity of using the possessive adjective ‘MY’ in relation to body to find out how the patients are concerned about their appearance.

Another point to consider in the text analysis was to investigate some characteristic words/tokens used by patients with cancer. As body image has a tremendous impact on patients due to the profound effect of the disease and the treatment on their appearance, and according to the authors, (Esplen and Fingeret, 2021 [9]) constitutes a critical psychological issue, we propose a list of words embracing body image that can support a treatment application tailored for special needs.

SAS Viya v.3,5, SAS Institute 820 SAS Campus Drive Cary, NC 27513-2414, USA, analytic software with open architecture was used for the analysis. We decided to use this software because it enabled us to mine, manage, modify, and retrieve data from a variety of sources and perform a statistical analysis. In addition, it provides comprehensive solutions for the part of speech tagging or categorizing the essential textual data. A two-step analysis was carried out. First, we incorporated the text corpus into the platform and used the following analysis nodes to retrieve morphological features: Text parsing, which provides us with Start/Stop list. In other words, it is the process of dropping and keeping the terms used in further analysis. Next points are stemming the terms to their basic form and part of speech tagging.Calculating the number of verbs and determining the verb tense (present or past), adjectives (categorize them into positive and negative groups), and the possessive adjective ‘MY’.Detecting characteristic terms referring to the oncological patient.

### 2.2. Sentiment-Based Approach in Body Image Analysis

The study uses a hybrid model approach, weighing the sentiment derived from the dictionary of emotions and the machine learning algorithm: deep recursive network. As the authors in the paper (Tao and Fang, 2020 [15]) notice, sentiment, as a part of NLP, traditionally classifies textual data as positive, neutral, or negative. We assumed the traditional model of sentiment based on terms from the general sentiment vocabulary, created by Wilson, Wiebe, and Hoffman, which is a database of nearly 8000 terms with specific polarity (positive, negative, or neutral) (Ali et al., 2013 [16]). In the dictionary-based part, the authors had to adjust the mentioned dictionary to the Polish language, as its original version was developed for English.

In the machine learning part, a deep recursive network with 5 hidden layers using LSTM (Long Short Term Memory) cells was used. LSTM is applied because it works well with long-sequence dependencies. As property vectors, we used 100-element pre-learned global property vectors developed according to the GloVe method. This method represents an unsupervised learning algorithm for learning vector representation of various terms (Huan et al., 2022 [17]). Due to the fact that our text corpus was relatively small when we compared it with the number of parameters used in an Artificial Neural Network-based model, we used the transfer learning method by using a pre-trained model using the Stanford Amazon Dataset (SAD). This dataset contains 34,686,770 reviews of product and user information, ratings, and a plaintext review (He, 2013 [18]). The mentioned method allows for accelerating the training process and improving the model performance by transferring previously learned features from one model to another. This is particularly useful in situations where the available dataset is relatively small (as in our case), which increases the risk of overfitting the model. Due to the relatively small number of patients from whom we collected the notes about body image, assembling a dataset similar in size and diversity to the SAD would be impractical or even unreachable in the context of our research. Therefore, using a pre-trained model on a large dataset is a beneficial approach. The model is fine-tuned on our own smaller, specialized dataset. As a result, despite the limited amount of data, satisfactory results have been achieved in recognizing features in patient notes.

### 2.3. Analysis of the Intensity of Basic Emotions

The developed analysis of the intensity of the 5 basic emotions, happiness, sadness, anger, fear, and disgust, is based on the extension of the Nencki Affective Word List (NAWL) dictionary. It is a standardized database of Polish words for studying emotions and other linguistic issues (Riegel et al., 2015 [19]). Words contained in the patients’ notes and not originally found in the dictionary (indefinite words) were added by interpolation. Emotion intensity was calculated in terms of distances to emotion cluster centers using the nearest-neighbors approach in the space of word embedding vectors. The solution was developed with SAS Viya and Python. A detailed description of the method is provided in (Barańska et al., 2022 [20]). The emotion intensity weights of the original dictionary were standardized according to the min–max formula: (1)$$f\left(x\right)=\frac{x-\min\left(x\right)}{\max\left(x\right)-\min\left(x\right)}$$

Base weights had a range from 1 to 7. Before the main steps for extending, some operations were performed for pre-processing, e.g., lower characters, deleting stop-words and special characters (numbers, punctuation marks), and finally, stemming. The latter process stands for transforming inflected words into their stems. Next, for notes with these word forms, we determined word embedding using a neural network. The dictionary extending process uses these numeric vectors to determine the emotion intensity of words which were not included in the original dictionary based on spatial position. This process consisted of 3 stages: 

Stage (I): finding the emotion center based on the dominant emotion of words coexisted in NAWL and notes. 

Stage (II): matching the indefinite word with the emotion center based on vector similarity using cosine similarity.

Stage (III): selecting the n-closest neighbors (from NAWL), which are the base for the emotion intensity. The calculation takes into account the distance between these words in space. The final formula for extending intensity is
(2)$$emo_{intensity}=\sum_{i=1}^{k}s_{k}\times w_{k}$$
where *s_k_* is the intensity of a certain emotion of k-word, and *w_k_* is the weight of a word based on the distance from the analyzed word.

A full dictionary allows us to present the emotional intensity of notes. It is calculated as the sum of the intensity of individual emotions multiplied by the number of words in the note for each emotion of the same individual emotions and finally divided by the length of the note (count in words).

### 2.4. Identifying Potential Areas of Difficulty

Potential areas of difficulty (topics) including body image, acceptance of the environment, and self-esteem were developed by a team of psychologists. In each area, we created dictionaries of representative vocabulary using the method of competency arbiters. As a method of processing documents with patients’ notes to detect whether a given topic was covered in them, we applied the method of detecting text topics based on the analysis of hidden dimensions.

A vector with the characteristics of a bag of words represented each note. Therefore, it was possible to determine how many times each word occurred in a given document. The final matrix in the rows contained information about individual words, and each column represented a given text document. Such a matrix is called a frequency matrix, which in geometric interpretation reflects a certain multidimensional space. In order to reduce the dimensionality of the obtained matrix, latent semantic analysis (LSA) was used, which is based on the singular value decomposition (SVD). Each dimension represents one of the topics, consisting of keywords. Each of these words is situated in a certain place in this space and has a certain value. A vocabulary analysis of each note allows us to determine the value of each note for each topic. 

Stage (I) included the preparation of data: notes and dictionaries. Text documents have been subjected to lemmatization, which consists of transforming words into their basic form. For each topic, two dictionaries were prepared: a dictionary of positive words and a dictionary of negative words depicting a given area. To check if the method used is correct, the notes were also shown to an expert psychologist to assess whether each document contains the analyzed area of difficulty. An expert psychologist marked each note by entering a specific flag (0—the topic was not raised in the note, −1—the topic was raised in a negative context, 1—the topic was raised in a positive context, 2—the topic was raised in a mixed context (positive–negative).

Stage (II) included the analysis to detect whether any of the analyzed areas of difficulty were raised in a given note.

Stage (III) included the analysis of the obtained results. Since a separate dictionary of positive and negative words was created for each difficulty area, in the results obtained for each note, the value of belonging to a given topic separately in a positive and negative context was known. For each of the notes and the topics, the relative difference was calculated according to the Formula (3).
(3)$$relative\;difference\;in\%=\frac{\left|W_{N}-W_{P}\right|}{\max\left(\left|W_{N}\right|;\left|W_{P}\right|\right)}\cdot 100\%$$
where *W_N_* denotes the value belonging to a given topic in the negative context (*N*), and *W_P_* denotes the value belonging to a given topic in the positive context (*P*).

Finally, we assumed that four types of results were possible: a topic is covered in a note in a negative context if the negative context topic affiliation value is greater than the positive context topic affiliation value and the relative difference is greater than 20%, a topic is addressed in a note in a positive context if the positive context topic affiliation value is greater than the negative context topic affiliation value and the relative difference is greater than 20%, the memo addresses a topic in a mixed context (to varying degrees of negative and positive context) if the relative difference between the affiliations is less than 20%, and a topic is not covered in a note if the value of belonging to that topic in each context is equal to 0. Then, the obtained results were verified with the expert flag.

### 2.5. The Research Material

The participants in the study were 50 patients (26 F, 24 M) with head and neck or upper gastrointestinal tract cancers staying in the gastroenterology unit of the oncological hospital in Warsaw, Poland. The mean age of the research group was 56, 84; SD 11,223. All patients included in the study stayed at the unit between August and December 2021. 

The inclusion criteria were as follows: diagnosis of H&NC or upper gastrointestinal tract cancer, enterally fed with active gastrointestinal access, underwent a surgery in the area of H&N or upper gastrointestinal tract in the past three months, were polish native speakers, remained in good verbal and logical contact, and gave informed consent to participate in the study. The qualified interviewer visited patients during their hospital stay and conducted an interview to gather basic anthropometric data. In the next step, the included patients were also asked the same question: “How do you perceive your body?” All patients were asked the same question, and, in case of doubt, the same detailed explanations were provided, i.e., “describe in your own words how you perceive your body, describe what your body looks like”. Patients were instructed to provide honest answers based on their current feelings while the interviewer took notes. Final notes were confirmed with the patients when they finished answering.

The consent of the bioethics committee to conduct the study was obtained by way of a resolution of 14 July 2021, ordinance 116/2018, issued by the Bioethics Committee of the Medical Center for Postgraduate Education in Warsaw, Poland.

## 3. Results

We performed a complex text analysis process to develop words statistics, structure, components, emotions, and potential areas of difficulties. The detailed process of obtaining the results is presented below (Figure 2). 

### 3.1. Statistical Analysis of Word Types and Sentiment Tagging

The analysis consisted of three parts. The first part included tagging and calculating parts of speech such as adjectives, verbs, and the possessive adjective MY regarding the patient’s body. Additionally, as for adjectives, the study was assumed to be detecting the total number of this part of speech tags and categorizing them with the positive and negative values. The verbs were analyzed in a similar way, but here, we also focused on tagging the past tense references. 

Figure 3 presents the general view of the analyzed objectives. The total number of adjectives is 191, of which 138 are negative and 53 are positive. Hence, we can state that patients present some level of body image concerns as they use more adjectives with negative connotations toward their bodies. The analysis of verbs showed that the overall number included 270 tags, where 105 stand for the verb “to be”. We registered 7 past forms of this verb, 89 referring to the present, and 9 to the future. The number of verbs related to states, actions, and verbs of perception is 163. In this group, there are 25 tags in the past form and 140 referring to the present. The overall number of the possessive adjective MY is 36.

In the next step, we developed a list of characteristic terms related to body image. Patients used plenty of negative words, mainly focusing on their face, skin, and hair, signaling concerns regarding the loss of their attractive appearance. They often referred to their bodies as ugly, old, dry, skinny, wrinkled, sick, devastated, weak, and disgusting. We also noticed that patients had concerns towards their skin, often characterized as dry, transparent, pale, rough, with rash, and allergic.

Bearing in mind that during the oncological treatment, patients suffer from hair loss, patients in their notes showed worries about their hair. They usually had difficulties with accepting hair loss and their fatal condition. Generally speaking, participants feel unattractive and mainly talk about death, pain, fear, and cachexia.

The analysis based on the hybrid model of sentiment resulted in 29 negative patients’ attitude towards their body, 10 positive, and 11 neutral.

### 3.2. Analysis of Basic Emotions Intensities

An expert in the field of psycho-oncology indicated the three main stages of the cancer disease process in terms of well-being and emotions dominant in the patients—diagnosis, treatment, convalescence, and palliative care. These stages can be linked to the Kübler-Ross model of mourning (Kübler-Ross, 1970). The model describes the series of emotions experienced by people who are grieving: denial, anger, bargaining, depression, and acceptance. In the first stage—the diagnosis—the patient can meet the difficulties related to beginning of the anticancer treatment, and it is often a traumatic experience. The emotional attitude of the patient is very unstable. Patients may experience a mixture of negative and difficult emotions. The patient may attempt to haggle and underestimate the diagnosis and, at the same time, manifest denial. Beginning the treatment can be understood as the second stage of the disease process. The patient’s emotional state stabilizes, and the dominant emotion is often anger. During the third stage of the disease—a convalescent patient is supposed to end the treatment and return to society. This stage might be associated with a sense of emptiness, lack of belonging, and mourning their healthy. We can also distinguish an additional stage of the disease process when the patient is redirected to palliative care. It is also called end-of-life care, and it coexists with the patients’ feelings of agreement. 

Figure 4 shows patients’ emotional profiles for the three main groups according to the stages—Stage I (diagnosis), Stage II (treatment), and Stage III (palliative care). Within the included cases, no one was in the convalescent stage. In the Student’s *t*-test and the Mann–Whitney U test, no significant differences were found. Statistically significant differences were found between the palliative group and stage II group (*p* = 0.0007) CI [0.056, 0.052] in the emotion of disgust and between the palliative group and stage I group (*p* = 0.003) CI [0.099, 0.084]. 

We included 11 patients in the Stage I group, 32 patients in the stage II group, and 7 patients in the palliative care group. In stage II, the level of anger was higher than in stage I. In the case of fear, the opposite occurs—the level of this emotion is higher in stage I than in stage II. The level of anger and fear was lower in the palliative treatment group than in the two remaining groups. The highest level of happiness was found in the group of patients treated in palliative care. The sadness was the highest in stage I and lowest in stage II. Disgust in stages I and II was on a similar level, and it was the lowest in palliative care. The results are in line with the psycho-oncologist professional’s opinion. It confirms that there is a difference in the emotional intensity depending on the disease progress. However, it was not statistically significant.

### 3.3. Identifying Potential Areas of Difficulties

The outcomes of the text mining-based assessment of challenging areas were compared to a psychologist’s expert evaluation. These comparative results are visually depicted in Figure 5. The chart illustrates the obtained level of agreement expressed as a percentage.

For the topic of body image, in the case of 39 notes, which is 78%, the result of the proposed method was consistent with the expert assessment. In the case of four notes (8%), a non-compliance error was noted. In the case of six notes (12%), despite the mixed context, the method detected one or showed a significant advantage of only one of the contexts. For one note, which is 2%, despite the specific context indicated by the psychologist, the computer method showed a mixed context. According to the expert opinion, among the notes with the mentioned topic, the share of notes with a negative context is 51%, with a positive context in 30%, and with a mixed context (negative–positive) in 19%.

For the topic of acceptance of the environment, for 41 notes, which represents 82%, the result of the proposed method was in line with the expert assessment. For four notes (16%), an error was noted. For one note (2%), despite the mixed context, the method detected one or showed a significant preference for only one of the contexts. According to the expert, among the notes with the mentioned topic, the share of notes with a negative context is 33% and with a mixed context (negative–positive) in 67%.

For the topic of self-assessment in the case of 40 notes, which is 80%, the result of the proposed method was consistent with the expert assessment. For two notes (4%), an error was noted. For eight notes (16%), despite the mixed context, the method detected one or showed a significant advantage of only one of the contexts. According to the opinion of the expert, among the notes with the mentioned topic, the share of notes with a negative context is 48%, with a positive context in 27%, and with a mixed context (negative–positive) in 25%.

## 4. Discussion

The developed method using NLP elements enables the quantitative determination of selected patient attributes. In the individual aspects of the proposed approach, dictionary methods, an artificial neural network, and the analysis of text topics, classified as classical methods, were used. The authors consciously made this choice, resigning from methods based, for example, on the so-called transformers due to the lack of a limited body size, for which collection was also part of the research. 

A large part of psychological research in oncology focuses on the patient’s well-being and quality of life. The proposed method tries to quantitatively analyze a specific research area: the perception of the body, which is an essential component of the patient’s quality of life (DeFrank et al., 2007 [21]). 

The obtained results of the intensity of emotions emphasize the difficulty of the end of treatment. According to Melissant HC et al., body-image disturbances among cancer patients are common and affect more commonly younger patients, or those after extensive surgery, and those who had wound healing problems (Melissant et al., 2021 [22]). Our findings are in line with those results presenting a great number of body image disturbances among the studied group. As presented in work by Jagannathan A et al., patients with a head and neck cancer diagnosis present a variety of feelings. The authors emphasized the need to evaluate and understand them, as they could be an important factor in cancer patient treatment. Patients experience great fear and a lack of identity, a sense of the fragility of happiness, and are often afraid to enjoy it. The solution presented in this study could support such a need and improve the quality of care.

In a review paper presented by Reich et al., they state that there is a need to assess the emotional status of oncological patients as well as immediately provide appropriate psychological care if needed (Reich et al., 2014 [23]). This study justifies the need to develop the tool presented in this study.

Our study highlighted the disturbances in body image perception. These results support the conclusion presented in the study by Cororve Fingeret et al. However, Fingeret et al. also stated that body image includes not only the visual parts of the body but also its functionalities (e.g., speech, swallowing) that were not assessed in our study (Cororve Fingeret et al., 2015 [24]). In another study of head and neck cancer patients, the main concern regarding body image reported by the patients were embarrassment and fear. In contrast to this result, our patients present mainly disgust or fear (Cororve Fingeret et al., 2012 [25]).

The important aspect of body image in head and neck cancer patients was stated by Henry et al., where the authors proved the reliability of a longitudinal study to assess and monitor body image. Our study included the measurement of body image disturbances only at one point in time and did not include observations (Henry et al., 2022 [26]).

As presented in the study by Macias et al., disturbed body image is related to distress and may impact the overall quality of life (Macias et al., 2021 [27]). However, our study did not include a deepened interview on patients characteristics to determine the factors most influencing the risk of body image disturbances.

Our study also did not show statistically significant differences in emotions present in patients answers, which remains in opposition to the thesis presented by Kubler-Rosa et al. This might be the result of the small study group and lack of an observation period. As stated in Covrig et al., there is a constant need for assessing body image disturbances in cancer patients (Covrig et al., 2021 [5]). Our method presented in this study demonstrates the feasibility of automating the process of assessing patients’ body image. However, it requires further refinement, including a more in-depth interviewing process, consideration of additional factors related to body image, and an extended observational period.

## 5. Conclusions

The amalgamation of a patient’s condition attributes, ascertainable through the specific method employed for that patient, can provide guidance on the appropriate support for the patient, their family members, direct medical staff, and psycho-oncologists. Due to the necessity of evaluating the quality of life and its components, particularly body image perception, among cancer patients, an automated method would facilitate healthcare professionals in swiftly conducting screenings in clinical settings. Further development of the method and its real-life testing in clinical settings is imperative.

### Limitation of the Work

The authors are aware of some limitations of the proposed approach. The main problem in the research was the fairly wide age range of participants, which explains the potential diversity of the vocabulary used.

Further research directions include gathering a larger research group and differentiating the results by gender, age categories, duration, and psychological phases of experiencing the disease. The vocabulary analysis showed specific areas worth further research, including skin, hair, bones, and vocabulary related to body devastation.

## Figures and Tables

**Figure 1 cancers-15-05437-f001:**
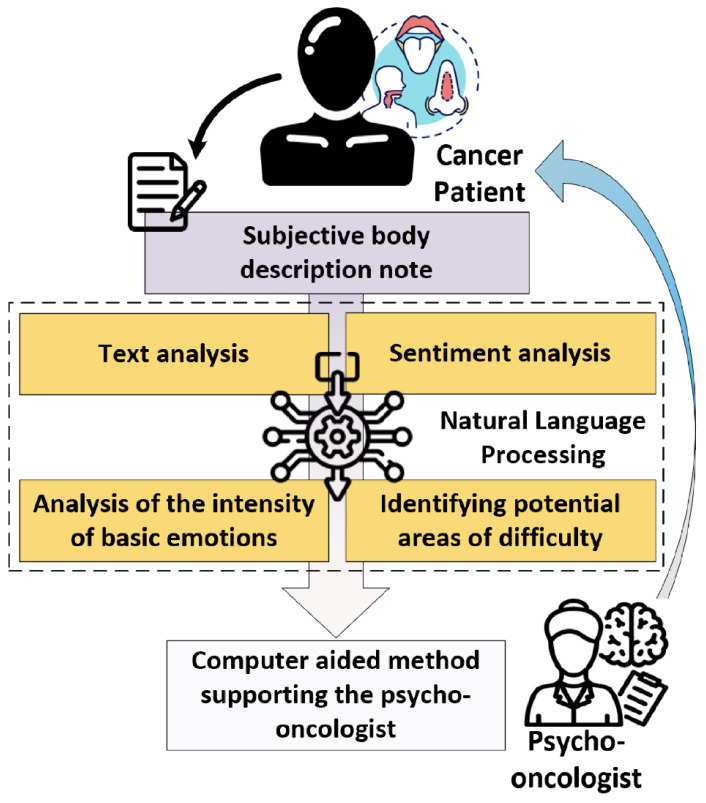
Summary of the elements of the developed method.

**Figure 2 cancers-15-05437-f002:**
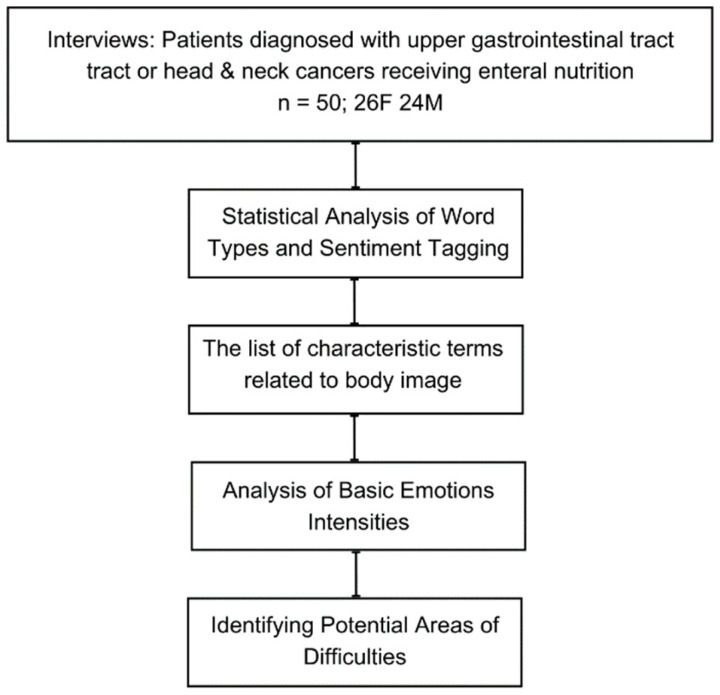
Schema of the planned medical experiment.

**Figure 3 cancers-15-05437-f003:**
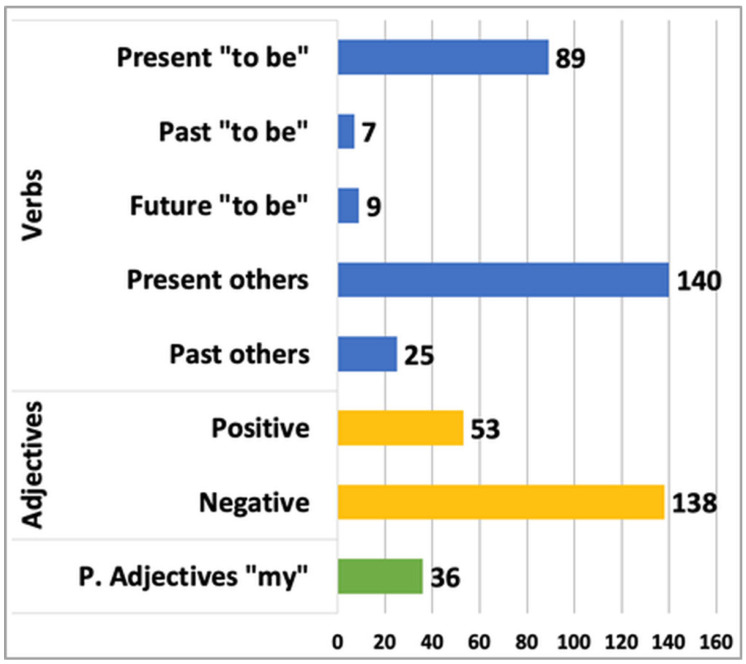
Results of statistical analysis of target words.

**Figure 4 cancers-15-05437-f004:**
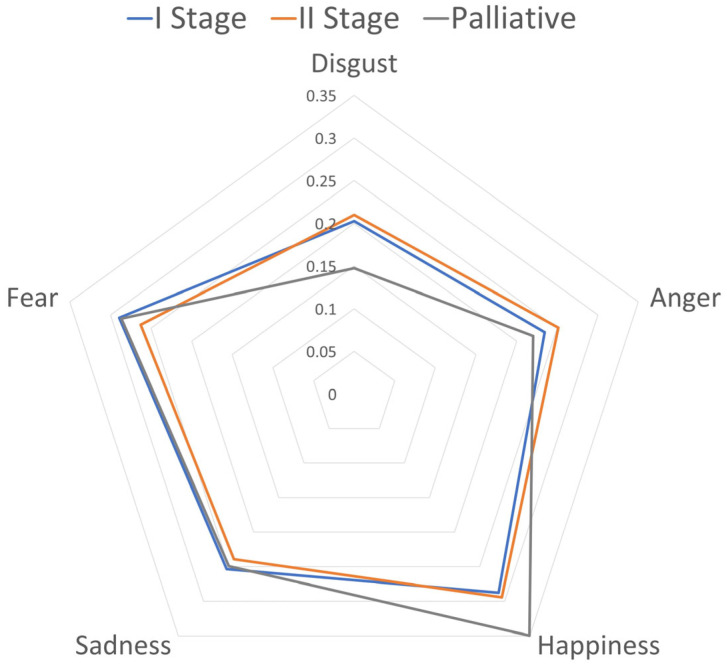
Emotional profiles for three staging groups—I Stage (diagnosis), II Stage (treatment), and Stage III (palliative care).

**Figure 5 cancers-15-05437-f005:**
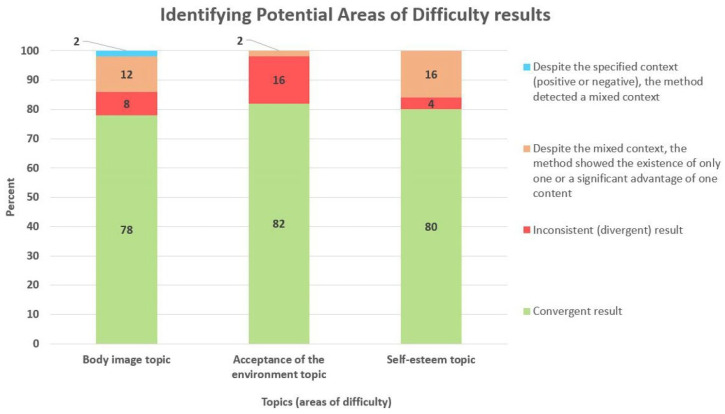
Verification of results achieved for the particular areas of difficulty: body image, acceptance of environment, and self-esteem.

## Data Availability

The data presented in this study are available on request from the corresponding author.

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
