# Peer review of "The Use of Natural Language Processing for Computer-Aided Diagnostics and Monitoring of Body Image Perception in Patients with Cancers"

_cancers, 2023, doi:10.3390/cancers15225437_

Round 1
Reviewer 1 Report
Comments and Suggestions for Authors
In this paper, the authors used a natural language processing (NLP) approach to perform body image analysis on patient note data. The proposed model can quantify patient attributes related to the psychological condition of patients, which may aid personalized healthcare. The addressed question is relevant and interesting to the medical informatics community. The reviewer has the following comments:
1. The authors trained the deep recursive network using the Stanford Amazon Dataset service data because their text corpus was relatively small. This may lead to training-testing set mismatch issues. An alternative approach is using transferring learning. For example, the authors can pre-train the model on the Stanford Amazon Dataset service dataset and use the pre-trained model as the starting point for training and tuning the model on their text corpus.
2. The Figure 3 results show that patients of different stages have different emotional profiles. The authors should perform statistical tests to check which differences are significant. This will shed light on which basic emotions are subject to the most changes when the patient's stage changes.
3. Figure 4 uses pie charts to demonstrate the convergent result proportions. Stacked bar charts may make the contrasts clearer for the readers.
4. (Minor) All the figures are blurry and should have higher resolution.
Author Response
- The authors trained the deep recursive network using the Stanford Amazon Dataset service data because their text corpus was relatively small. This may lead to training-testing set mismatch issues. An alternative approach is using transferring learning. For example, the authors can pre-train the model on the Stanford Amazon Dataset service dataset and use the pre-trained model as the starting point for training and tuning the model on their text corpus.
Thank you for that impactful comment. The solution you suggested was indeed implemented by us. We have also included additional descriptions in the text body to provide a more thorough explanation.
- The Figure 3 results show that patients of different stages have different emotional profiles. The authors should perform statistical tests to check which differences are significant. This will shed light on which basic emotions are subject to the most changes when the patient's stage changes.
Thank you for this comment. We performed the appropriate tests and added description in the text. See lines 339-343, 423-430.
- Figure 4 uses pie charts to demonstrate the convergent result proportions. Stacked bar charts may make the contrasts clearer for the readers.
Thank you for your kind suggestion. We implemented your comment and changed the chart type. See Figure 4.
- (Minor) All the figures are blurry and should have higher resolution
We appreciate your keen attention to detail. We have taken the initiative to provide higher-quality images, which can be found in a separate file attached to the manuscript.
Reviewer 2 Report
Comments and Suggestions for Authors
The manuscript made by Gliwska E et al. is interesting and well done; the methodology is extensive, well explained, and informative. I think it will be reproducible for authors interested in this topic.
Figure 1 is detailed; they explain in a summary form the methodology used. The well-made methodology is closely related to the interpretation of the results that were well explained; the results are extensive and resolve the variables studied.
The discussion provides important data that will be functional for further studies. The quality of the discussion is the result of the well-made methodology and the well-explained results.
Thank you to the authors for explaining the limitations of the study and the purpose of further studies.
Author Response
Thank you for your positive opinion about the submitted manuscript.
Reviewer 3 Report
Comments and Suggestions for Authors
Overall, the manuscript “The use of NLP elements for computer-aided diagnostics and monitoring of body image perception in patients with head and neck or of the upper gastrointestinal tract cancers” provides a comprehensive and detailed description of the research methods and results. However, there are some areas that could be improved to enhance the clarity and quality of the manuscript. Here are some suggestions for revision:
1. The manuscript could benefit from a more descriptive title that clearly indicates the research focus and the use of Natural Language Processing (NLP) techniques. A more specific and informative title would help potential readers understand the content briefly.
2. In the introduction, it's essential to provide a clear and concise statement of the research objectives or hypotheses. What specific questions or problems is this study addressing? The introduction should also briefly mention the significance of the study and how it contributes to the existing body of knowledge. Why is it important to analyse patients' body image in the context of oncology treatment?
3. The "Materials and Methods" section should start with a clear explanation of the research design and data collection process. Readers should understand how the study was conducted before delving into specific analysis methods. For each method or technique used (e.g., text analysis, sentiment analysis, emotion intensity analysis), provide more context and explanations. Explain the rationale behind using these methods in this specific study.
4. The results section should be organized in a more structured manner. Present the key findings in a logical order and consider using subsections to separate different aspects of the results.
5. In the discussion, relate the findings back to the research objectives and the existing literature. Explain the implications of the results and how they contribute to the field of psycho-oncology.
6. The manuscript briefly mentions limitations, such as the wide age range of participants, but it would be helpful to elaborate on these limitations and their potential impact on the results. Discuss potential directions for future research in more detail, including how these limitations could be addressed in future studies.
7. The manuscript would benefit from improved clarity and organization. Make sure the text flows smoothly and is well-structured. Avoid excessive jargon or technical terminology without adequate explanations. Ensure that the text is accessible to a broad audience, including those not familiar with NLP.
By addressing these suggestions, the manuscript can be improved to make the research more accessible, and its findings more easily understood by a wider audience.
Comments on the Quality of English Language
Minor English refinement is required by a native speaker.
Author Response
1.The manuscript could benefit from a more descriptive title that clearly indicates the research focus and the use of Natural Language Processing (NLP) techniques. A more specific and informative title would help potential readers understand the content briefly.
2.In the introduction, it's essential to provide a clear and concise statement of the research objectives or hypotheses. What specific questions or problems is this study addressing? The introduction should also briefly mention the significance of the study and how it contributes to the existing body of knowledge. Why is it important to analyse patients' body image in the context of oncology treatment?
Thank you for this insightful comment. We have incorporated the suggested change to the manuscript title and provided a more detailed explanation of the study's aim in the introduction, as per your recommendation. You can find these revisions in lines 67-72.
3. The "Materials and Methods" section should start with a clear explanation of the research design and data collection process. Readers should understand how the study was conducted before delving into specific analysis methods. For each method or technique used (e.g., text analysis, sentiment analysis, emotion intensity analysis), provide more context and explanations. Explain the rationale behind using these methods in this specific study.
Thank you for that insightfull comment. We addressed your request and added more explanation in lines 267-272 and we have added the Figure 2 to explain the schema of the experiment.
4. The results section should be organized in a more structured manner. Present the key findings in a logical order and consider using subsections to separate different aspects of the results.
Thank you for your comment. We have adjusted the results section according to your advice and prepared a graph illustrating the process of obtaining results. We believe that the results section is now clearer.
5. In the discussion, relate the findings back to the research objectives and the existing literature. Explain the implications of the results and how they contribute to the field of psycho-oncology.
We appreciate your comment and reorganized the discussion.
6.The manuscript briefly mentions limitations, such as the wide age range of participants, but it would be helpful to elaborate on these limitations and their potential impact on the results. Discuss potential directions for future research in more detail, including how these limitations could be addressed in future studies.
Thank you for this comment. We appreciate your thoughts and included more descriptions of the limitation in the discussion part.
7.The manuscript would benefit from improved clarity and organization. Make sure the text flows smoothly and is well-structured. Avoid excessive jargon or technical terminology without adequate explanations. Ensure that the text is accessible to a broad audience, including those not familiar with NLP.
After adding additional information, we assume that the article will be more understandable to a wider group of readers.
By addressing these suggestions, the manuscript can be improved to make the research more accessible, and its findings more easily understood by a wider audience.
Round 2
Reviewer 3 Report
Comments and Suggestions for Authors
Dear authors, thank you for addressing the comments, I have no further points to discuss.